# Investment in Forest Watershed—A Model of Good Practice for Sustainable Development of Ecosystems

**Iulia Diana Arion** [1] , **Felix H. Arion** [2,*] , **Ioan Tăut** [1,3,*], **Iulia Cristina Mureşan** [2] , **Marioara Ilea** [2] and **Marcel Dîrja** [4]

1. Department of Forestry, Faculty of Forestry and Cadaster, University of Agricultural Sciences and Veterinary Medicine of Cluj-Napoca, 3-5 Calea Mănăștur St., 400372 Cluj-Napoca, Romania
2. Department of Economic Sciences, Faculty of Horticulture and Business in Rural Development, University of Agricultural Sciences and Veterinary Medicine of Cluj-Napoca, 3-5 Calea Mănăștur St., 400372 Cluj-Napoca, Romania
3. National Institute for Research and Development in Forestry (INCDS) "Marin Drăcea", Cluj Branch, Street Horea, no. 65, 400202 Cluj Napoca, Romania
4. Department of Terrestrial Measurements and Exact Sciences, Faculty of Forestry and Cadaster, University of Agricultural Sciences and Veterinary Medicine of Cluj-Napoca, 3-5 Calea Mănăștur St., 400372 Cluj-Napoca, Romania
* Correspondence: felixarion@usamvcluj.ro (F.H.A.); ioan.taut@usamvcluj.ro (I.T.)

**Abstract:** Estimating the efficiency of the investments that generate public benefits is not an easy task, as there are various methods proposed for evaluating the value of public goods and services. The novelty of the study consists of the proposed cost–benefit methodology adapted to a real-value cost concept for estimating the efficiency of the investment in the hydrographic Fâncel watershed, in the center of Romania, using a set of five indicators. The results prove that an investment of RON 1,323,226.29, at the value of 2018, would be paid in 5 years, 0 months, and 15 days. The estimated income is RON 505,327.8/year, so at a 5% financial-discount rate and an estimated period of use of 30 years, the net present value is RON 5,612,730.67. The internal rate of return was calculated as 22%, whereas the value of the savings-to-investment ratio is 3.6%. The main findings of the analysis of the economic-efficiency indicators offer a synthetic and relevant image of the fact that the investment has proven to be effective under the analyzed conditions. The results offer arguments to assume that the decision to consider that particular investment a good practice is fulfilled.

**Keywords:** hydrographic; torrent correction; efficiency; investment

## 1. Introduction

It is widely accepted that forest ecosystems are suffering a level of degradation at the moment [1], which is complicated to evaluate [2,3]. This is a result of the fact that different countries use various methods of phenomenon estimation and also because degradation is caused by miscellaneous factors: human intervention, climate events, pests, fire, and diseases, to mention a few. The effects of forest degradation are complex [4] and have to be prevented and corrected.

The resolution adopted by the General Assembly of the United Nations (UN) as Transforming our World: the 2030 Agenda for Sustainable Development [5] includes 17 goals. Goal 15 is dedicated to forestry, including the sustainable use and management of terrestrial ecosystems. Among other aspects, it is stipulated that the use of substantial funds by 2023 is needed in order to conserve freshwater ecosystems, prevent degradation, and fight against floods. A study [6] regarding the impact of big data on these goals revealed that Goal 15 is closely related to Goal 13 (climate action) and Goal 6 (clean water) in terms of assuring sustainable water ecosystems. Furthermore, the paper proves the relevance of this particular research to other studies on water-resource management, land degradation,

and natural disasters. Other studies [7] have proven that the degradation reported by Goal 15 of the UN is mainly a result of human activities and climate factors [8].

Some authors [9] even consider that the above-mentioned goals must be more orientated to assure all three pillars of sustainability, focusing not only on the social one but also on the economic and environmental ones. For instance, for appropriate management of forest concessions, a set of voluntary guidelines [10] has been designed summarizing eight principles, one of which stipulates the necessity of assuring long-term economic and financial sustainability.

The main objective of the work is to evaluate the investment in the small hydrographic watershed Fâncel, in the center of Romania. The evaluation could be used as a model of good practice for sustainable development as an integral part of the integrated water–forest strategy for the area. In this way, it contributes to the understanding of investment in the hydrographic watershed in general. The study addresses a knowledge gap in the methodology of effective investments in infrastructures that deal with natural factors, offering a new perspective on the subject. To do so, the economic efficiency of the hydrographic watershed, based on an adapted cost–benefit methodology with a real-value cost concept, was estimated. The work is multifaceted, based on the complexity of the estimation of the potential cost and revenues, which implies expressing units of the various types of non-financial goods (e.g., environmental, social,). The secondary objective is to draw attention to the importance of an in-depth analysis of the economic efficiency of investments in watersheds.

## 2. Literature Review

Nevertheless, to organize forest management in a sustainable way, a unique human involvement component is required, which has to be correctly dimensioned so as to cope not only with the environmental situation [11–13] but also with the social and economic expectations [14–16]. Researchers [17] have found that the results of diverse policies for stopping the phenomenon of deforestation do not function as expected if the allocation criteria are not adequately correlated with the geographical specificity of the area [18] and with the socio-economic realities. Disruptions in forests are a serious situation that generates an accumulation of sediments, disrupting the regulatory function of water in the forest [19]. From this point of view, forest-water resource management should be closely connected to the systems of human needs [20].

At the moment, no plan for sustainable forest management can neglect water as an indispensable segment [11]. Within this integrated vision, the issue of water in the forest-management process is determined by the value of water. Its value is either known as a market price or estimated on the basis of economic assessments. Either way, water-management policies [21] should benefit from the relevant information in order to provide viable solutions for better management of this natural resource. Using a unique value for water is not a viable option. The basic concept considers that a need is not fixed in time and, therefore, the water needs depend on the amounts of water used at a certain moment. Consequently, its total and marginal economic value can oscillate [22]. The economic value of water used for agriculture and industry [23] depends on the marginal residual value of water for production. In the case of urban supply and of other end users [24], its value depends on the willingness to pay for it.

In the near future, water-resource management will include the design and construction of new water-supply systems in parallel with a better functioning of the existing ones. The value of water is not fixed, and that involves additional efforts to find solutions to water deficiencies and reduce water conflicts. As a result, attention will be paid to the hydro-economic models, which are systems developed in an integrated manner focusing on water-resource distribution and on the existing infrastructure, with management options based on economic value [25].

The economic view has become more integrated with the traditional engineering and hydrological models of water-management issues. It is required to combine economic-

management concepts and performance indicators with technical ones. In this manner, the hydrological system can offer more useable information for deciding on water management and policies [14,26,27]. When such models are developed and used with the involvement of the interested parties, they can become a basis for the common understanding of water issues and a keystone for management solutions and accepted public policies [24]. It has been stated [28] that these combined hydro-economic models of watershed are the key tools for evaluating the strategies that are created to improve the economic efficiency of water use in a context of competition for limited water resources.

Integrated hydrological research, which combines the social and natural sciences, can partly help to address the current failure to provide relevant information for achieving the necessary coherence in different government policies. An integrated river-basin research framework suggests [29] that a combination of economic assessment, integrated modeling, stakeholder analysis, and multi-criteria assessment can provide complementary perspectives on wetland management and on welfare-optimization policy. Subsequently, each of the different components of such integrated wetland research is reviewed and linked to the wetland-management policy. The environmental effects caused by hydro-technical constructions are numerous and profound, both positively and negatively. They should be analyzed using complex tools, including economic-science methods [30–32], in order to estimate and control the impact of anthropogenic activity on the environment: on the health and safety of fauna, flora, soil, air, water, climate, landscape, and historical monuments or other constructions.

The Bonn Challenge Barometer and indicators are an output of The International Union for Conservation of Nature, which created a specific barometer at the disposal of countries that are involved in the Bonn Challenge initiative. The purpose is to create a unitary and flexible approach to forest and landscape restoration [33]. The proposed methodology includes inputs (success factors, results, and benefits) and outputs (progress, financial, technical, policy, government constraints, and opportunities) [34].

Successful stories are relevant and the use of proper estimation and of funds for investment has been assessed as well. Nevertheless, governments and managers have to invest in reducing degradation, including degradation generated by flood risk. However, governmental resources are limited [35] and the needs are various [36], so a cost–benefit analysis could be one of the criteria for selecting the investments. The Nature Conservatory along with Agence Française de Développement designed a practical guide [37] for watershed investments in which it is recommended to use cost–benefit criteria compared to the benefits returned by other potential investment objectives. There are studies [38] that have proposed viable solutions for ranking the potential investment projects intended for forest protection and restoration.

The assumption is sustained by the Food and Agricultural Organization of the United Nations (FAO), which clearly stipulates that forests have a reliable prospective to deal with the hazards the world is confronting today. Nonetheless, the interventions have to be rapidly scaled up, available wherever needed, put into effect without delay, and cost effective [39]. In the United States (US), the Forest Service Manuals of the Forest Service of the US Department of Agriculture (Part 2500—Watershed and Air Management, Chapter 2520—Watershed Protection and Management, 2521.11—Use of Watershed Condition Classes, 2521.11a—Needs Identification and 2521.11b—Priority Setting) [40] stipulates that when targeting watershed priorities, establishing criteria in terms of economic feasibility has to be considered, together with the technical feasibility and other resource factors—including values and benefits, economic effects, social factors, and potential partnerships.

Forest management, which has to be viewed as a tool not only for producing timber but also for assuring sustainable development, has generated increased socio-economic attractiveness [41]. Scholars have also mentioned that an appropriate balance of economy and ecology should be the final goal of the improvement process of the environmental aspects of forests [35,42–44]. The reasons behind this consist of the fact that the sustainability of forests was and still is approached independently in terms of each pillar of sustainability,

and less in terms of the relations between the ecological and socio-economic pillars [45]. The creation of balanced socio-economic and ecological development is widely known as an issue with no obvious answer [46]. To achieve this goal, some [47] advocate for setting up new institutions: firstly, localized, participatory decision-making to establish and plan the activities, and secondly, a financial mechanism to collect funds for implementing the activities.

Further studies have confirmed [48] that the management of natural resources, including water resources [49] in forest areas, demands appropriate integration into the holistic local strategy developed with the support of stakeholders [50,51] and locals [52]. Growing experience with participatory forest management has also been observed [53]. The concept of a localized approach has been confirmed by other scholars who consider that the watershed externalities cannot be managed effectively if carried out at a centralized level [54].

## 3. Materials and Methods

The study was run on the small Fâncel watershed (Figure 1), part of the Gurghiu watershed, in the forest fund of Fâncel, Mureș County (Romania). The Gurghiu watershed is situated in the eastern part of the Transylvania Depression. It is positioned in the northwest part of Mureș County in the hydrographic basin of the Mureș River (Figure 1).

The Gurghiu hydrographic basin includes 16 hydrographic subbasins with areas between 9.43 and 206.86 km$^2$. Based on the size of the torrential hydrographic basins, two subbasins are medium, with an area of 943 ha (Mocirlosul) and 916 ha (Păuloaia); 13 subbasins are large, with an area of up to 10,000 ha; and the Gurghiu sub-basin, with an area of 20,686 ha, is a very large subbasin. The Fâncel subbasin has a circularity coefficient of 1.95 and tends towards an elongated basin. This is due to the geological diversity and to the different resistance to erosion conditioned by the local base levels.

From a hydrological point of view, it can be concluded that the main stream has a relatively constant flow during the year. The tributaries benefit from variable flows, especially after heavy rains and snow melt. Among the sources of superficial feeding, rain has the largest share, at 60–70%. Snow is less abundant, with a share of 30–40%. There is an increase in debits in March, with the thawing and melting of the snow, which extends until May due to the spring rains. High flows occur in the summer months due to short-lived torrential rains, followed by increases in flows in November and decreases during winter. The highest flows are caused by overlapping of the snowmelt period with the spring rain.

Average annual temperature variation is in the range of 1.4–8.90 °C. The annual average data highlight the fact that the warmest months are May, at 15.80 °C; June at 19.10 °C; July, at 21.10 °C; August, at 20.50 °C; and September, at 15.60 °C. The variation in precipitation is in the range of 685.7–1300.8 mm/year, with a maximum in June–July and a minimum in January and March, which shows the continental nature of the precipitation.

The longitudinal profile of the valleys has a slope of 5–6%, but on their tributaries the slope is much steeper, at 25–30%. The torrential nature of the valley is obvious, manifesting with intensity in the floods, with power of erosion and high transport of alluviums (Figure 2). The total surface of the watershed of these torrential formations is 6527.8 ha, which is divided into 6460.7 ha of forest fund and 67.1 ha of agricultural fund [55].

Precipitation in the area is more abundant at the end of spring and summer and less abundant in autumn and winter. The annual average number of days with precipitation is 145 and that of days with snow is 50. The average number of days with snow cover is 100 (Table 1).

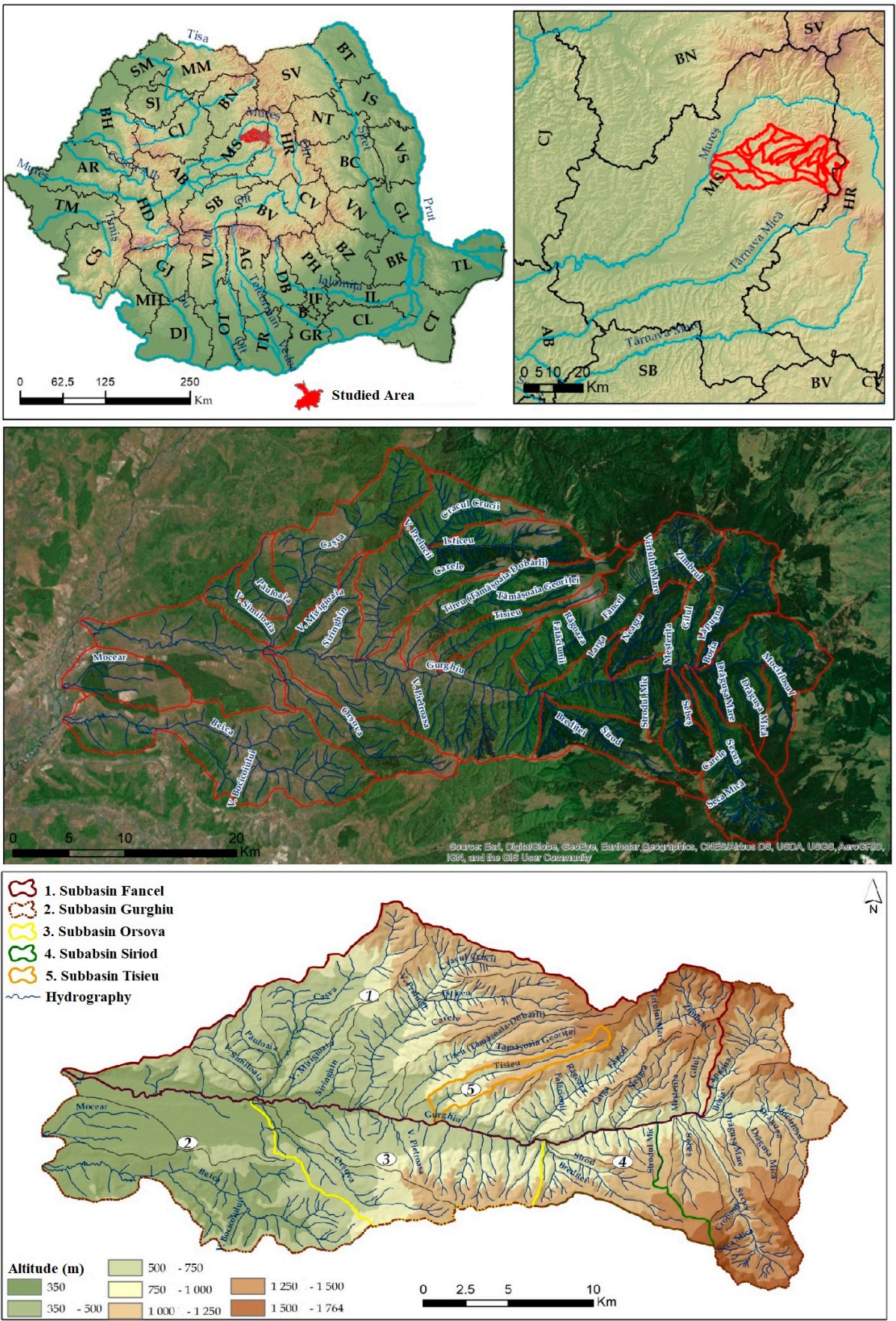

**Figure 1.** Geographical location of the Gurghiu watershed.

**Table 1.** Rainfall regime (mm) between 2015 and 2018.

| | Month | | | | | | | | | | | | Annual Average |
|---|---|---|---|---|---|---|---|---|---|---|---|---|---|
| | January | February | March | April | May | June | July | August | September | October | November | December | |
| Normal value in 50 years | 26.7 | 24.1 | 28.6 | 48.7 | 72.2 | 88.2 | 78.3 | 62.9 | 49.8 | 41.8 | 34.3 | 32.9 | 588.59 |
| Monthly sum 2015 | 22.9 | 28.3 | 26.8 | 35.8 | 67.6 | 127.4 | 32.1 | 78.8 | 72.7 | 37.8 | 36.5 | 11.3 | 578.0 |
| Observation | Dry | Little rain | Normal | Dry | Normal | Very rainy | Very dry | Rainy | Very rainy | Normal | Normal | Excessively dry | Normal |
| Monthly average 2016 | 36.6 | 28.2 | 38.0 | 71.8 | 71.2 | 76.2 | 77.4 | 71.4 | 17.8 | 75.6 | 52.6 | 19.9 | 636.7 |
| Observation | Very rainy | Little rain | Very rainy | Very rainy | Normal | Slightly dry | Normal | Little rain | Excessive rain | Excessive rain | Excessive rain | Very dry | Little rain |
| Monthly average 2017 | 12.5 | 24.9 | 23.0 | 42.2 | 127.6 | 82.4 | 80.4 | 29.6 | 43.0 | 61.7 | 48.6 | 38.9 | 614.8 |
| Observation | Excessively dry | Normal | Slightly dry | Slightly dry | Excessive rain | Normal | Normal | Excessively dry | Slightly dry | Very rainy | Very rainy | Little rain | Normal |

Note: Source: National Administration of Meteorology in Romania.

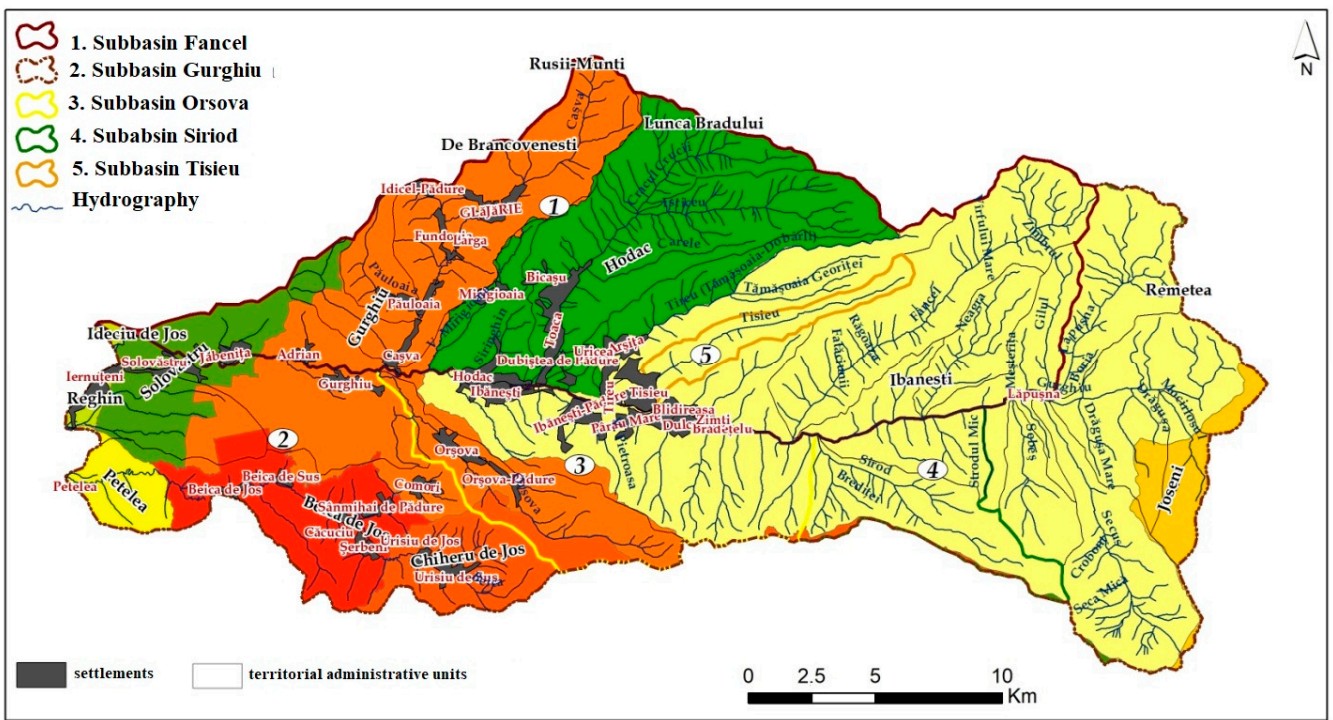

**Figure 2.** Geographical location of the study area—small Fâncel watershed.

### 3.1. Adopted Method

In order to determine the choices that maximize the benefits and suppose a monetary expression of all direct, indirect, and intangible effects, it is necessary to apply the cost–benefit analysis (CBA), which supposes the comparison of the cost of a project with the benefits that it can bring. The used methodology is an adaptation of the recommendations of The World Bank Group [56], which emphasizes the importance of economic and financial analysis of the establishment of a process- for watershed investment. It takes into consideration inputs and outputs, including externalities and "cost of inaction" [57], expressed as real-value costs.

Studies prove that a compromise between the cost of an investment and the results should be achieved, expressed both qualitatively and quantitatively [58]. Thus, the situation after the investment should be compared with the one when an investment was not made. Therefore, the correct setting of the real level for comparing references accounts for the key factor to express the cost–benefit for an investment in ecosystems [59]. Various studies [60,61] have examined the effects of using the concept of the economic efficiency of investments in torrential-watershed development, created to regenerate the natural potential. They expose the differences between the cases with and without investment in monetary units. In practice, there are more ways of analyzing both economic costs and benefits in the process of decision-making for watershed investments and maintenance [62], adapted to the local specificity.

It can be observed that the methodology is approached in different ways. There are scholars [63] who advocate that a cost-effective strategy and mainstreaming investments in watershed services require the development of practical tools for cost–benefit measurement. Consequently, the dissemination of good practices should be replicable both institutionally and financially.

### 3.2. Cost and Revenue Estimation

The literature systematized the method of estimating the cost, revenue, and value of the avoided potential losses [64]. Thus, the total cost of the arrangement of the torrential watershed represents the totality of the amount spent in order to execute the totality of

the categories of work required for the elimination of erosion and torrential processes. The costs are represented by the expenses due to the afforestation of the degraded lands, consolidation works on the slopes, hydrographic network-consolidation works, completion, maintenance and repair works, and more. The estimation of expenses was made according to the projects and norms in force. In this sense, the results of the research were used to substantiate and develop various plans to manage flood risk.

The degraded hydrographic network is both longitudinally and transversely consolidated by such torrent-correction works. The longitudinal works were as follows: stone-masonry channels designed to take the water and drive it to the forest-road bridge paths, thus preventing the erosion from spreading or clogging and preventing the interruption of traffic; and stonewall support walls, located at the base of the slope and designed to support the embankment slopes of forest roads. Transversal works included thresholds and dams made of stone masonry built to retain alluvium; to consolidate riverbeds, banks, and the slopes of the roads in the neighborhoods; and to mitigate floods and regulate leakage.

The following stages were taken into account for the cost–benefit analysis: delimitation of the study, analysis of objectives and actors, identification of protection strategies, analysis of the space, assessment of the precision of the ladder, identification and evaluation of costs and benefits, evaluation of the costs of the protection works, assessment of the benefits (potential damages), choosing an update rate, comparison of costs and benefits, choosing decision criteria, and calculation of the following economic-efficiency indicators: specific investment (I), investment-recovery duration (D), net present value (VNA), internal return rate (RIR), and savings-to-investment ratio (IR).

*3.3. Indicators*

The total investment value was updated at the level of the year 2018 by observing the recommendations included in [12,13], which were also implemented by the Romanian legislation [65]. Therefore, it was calculated as follows:

$$VA = \frac{I}{(1+r)^t} \tag{1}$$

where I = investment; r = update rate; t = year for which the update is operated.

In order to analyze the economic efficiency of the torrent-correction works within the small Fâncel watershed, a number of specific indicators were used:

1. Investment (I)

   The investment represents the value of the invested capital.

2. Payback Period (D)

   The investment-recovery duration is in fact the time required to recover the invested capital and is calculated by dividing the value of the investment by the amount of net cash inflow that the project generates each year.

3. The Net Present Value (VAN)

   The net present value is the updated value (based on a specific update rate) of all income and costs of the investment, calculated as the difference between the updated annual revenue and the updated investment value:

$$VAN = \sum_{n=0}^{t} \frac{(Vt - Ct)}{(1+r)^t} \tag{2}$$

where Vt = income of year t; Ct = costs in year t; r = discount rate; t = calculation year.

4. Internal Rate of Return (RIR)

The internal rate of return expresses the amount of the update rate that is required for up-to-date income to be equal to the updated cost—namely, the update rate for which the updated net value is equal to 0.

The internal rate of return is calculated using the following formula:

$$0 = \sum \frac{(Vt - Ct)}{(1 + RIR)^t} \tag{3}$$

where Vt = income from year t; Ct = costs in year t; RIR = internal return rate; t = calculation year.

5.  Savings-to-Investment Ratio (IR)

The savings-to-investment ratio is the ratio of the updated value of the savings to the updated value of the investment, being the result of the following formula:

$$IR = \frac{VAN + I}{I} = \frac{VAN}{I} \tag{4}$$

VAN = net present value; I = investment in year t.

The value of the costs and revenues incurred by the hydrological developments within the small Fâncel watershed were subsequently updated to the year 2018 by the updated cash-flow method using an update rate of 5%, according to the National Guidelines for the cost–benefit analysis of projects funded by structural instruments, developed by the Authority for the Structural Instruments Coordination within the Ministry of Economy and Finances [65]. The costs of the related works were taken from the specific project documentation for the construction works, developed during the year 2007.

In terms of income, it was estimated both in the forestry sector (recovery of timber, recovery of berries, recovery of honey, recovery of resin, recovery of other products—medicinal plants, leaves, cones, bark, etc.) and in other sectors (agriculture and animal husbandry—reduction of soil erosion, tourism, spa treatment, water accumulations—energy, irrigation, household consumption, etc.) [66]. The use of the true-value accounting concept that estimates the natural and social impacts on the environment is promoted by the FAO [67] and scholars in diverse areas [68–71]. In that way, the importance of financially expressing the value of ecosystem services is acknowledged. As a result, the decision-making process will be more accurate. There is a number of available techniques [72] for measuring the value of such services so they can be consolidated into decision-making.

A method meant to estimate the costs, income, and value of the potentially avoided loss has been proposed [73]. Thus, the total cost of the improvement of the torrential watershed represents the total amount spent to execute the total categories of works that are required to eliminate erosion and torrential processes. Within certain research works developed in the period 2000–2002, some advanced technologies for hydrological and anti-erosion improvements of the torrential watershed were identified [74]. They assume that economic and technical efficiency is affected by a series of factors, including the watershed morpho-dynamics, their evolution over time, the production obtained, and the nature and magnitude of torrential and degradation processes.

The estimation of the value of the damages and/or losses that have been avoided by arranging the torrential watersheds was carried out by evaluating the value of the potential losses related to the destruction of the soil-production capacity, losses related to the failure to carry out the arrangement works on time, losses caused by torrential alluvial processes, direct damage caused by torrential floods and landslides, and damage related to the decommissioning of objectives for a period of time. The currency used in this paper is RON (Romanian new leu), and the exchange rate was, as an average during the year 2018, RON 4.6540 to EUR 1 [75].

## 4. Results and Discussion

### 4.1. Reasons for Investment

The main objective of the investment in torrent correction in this particular small watershed was to protect the forest roads of the area against consequences of torrential rain and to avoid frequent disruptions to the circulation of the exploitation of the wood material of the watershed. In addition to this main objective, other objectives in the area were to protect other natural and anthropic assets, such as human households, forest and agricultural land, the riverbed of the Mureş River, and downstream rural localities. Of the listed objectives, the most vulnerable are the forest roads. The hydrographic network-development works help to strengthen alluvial deposits and slurry cones, degraded slopes, and unstable shores; to redirect the floodwaters without causing erosion of banks and slopes; and to partially retain the transported material.

### 4.2. Estimation of Costs

The total investment was valued at RON 2,263,166 (Table 2) and was updated to the level of year 2018 by the recommendations included in the Guide to Cost–Benefit Analysis of Investment Projects—Economic Appraisal Tool for Cohesion Policy 2014–2020, elaborated by DG REGIO—Directorate-General for Regional and Urban Policy [76], which was also implemented by the Romanian legislation [65].

**Table 2.** General cost estimation of torrent-correction works.

| No. | Chapters and Sub-Chapters of Costs | Total Values (RON, VAT Included) |
|---|---|---|
| | CHAPTER I | |
| | Expenses for Obtaining and Arranging Land | |
| 1.1 | Obtaining Land | 0 |
| 1.2 | Landscaping | 0 |
| 1.3 | Environmental Protection Facilities | 0 |
| | Total Chapter I | 0 |
| | CHAPTER II | |
| | Expenditure on Design and Technical Assistance | |
| 2.1 | Field Studies | 2380 |
| 2.2 | Obtaining Advises, Agreements, and Authorizations | 20,651 |
| 2.3 | Design and Engineering | 81,277 |
| 2.4 | Organization of Public Procurement Procedures | 4380 |
| 2.5 | Technical Assistance | 28,560 |
| | Total Chapter II | 137,248 |
| | CHAPTER III | |
| | Expenditure on Basic Investment | |
| 3.1 | Building and Installation Objective Pr. NEGRU | 1,068,24 |
| 3.2 | Building and Installation Objective Pr. FÂNCEL | 589,863 |
| 3.3 | Building and Installation Objective Pr. FĂTĂCIUNIŢA | 257,134 |
| | Total Chapter III | 1,915,23 |
| | CHAPTER IV | |
| | Other Expenses | |
| 4.1 | Organization of Site | 86,186 |
| | 4.1.1. Construction Work and Related Installations | 68,949 |
| | 4.1.2. Costs Related to the Organization of the Yard | 17,237 |
| 4.2 | Fees, Taxes, Legal Fees, Financing Costs | 21,864 |
| | 4.2.1. Commissions, Fees, and Legal Fees | 21,864 |
| | 4.2.2. Credit Cost | 0 |
| | Total Chapter IV | 210,675 |
| | TOTAL GENERAL | 2,263,166 |

Note: Source: Preliminary studies carried out to substantiate the need for investment within the small Fâncel watershed [55].

Like any investment project, the outflows were discounted by means of a financial discount rate (5% [65]) using Formula (1), so the investment of RON 2,263,166 from 2007 was evaluated as RON 1,323,226.29 RON in the year 2018.

Based on the national approval of the catalogue for the fixed-assets classification and normal operating durations [77], assets included in the Dam category (defense, subdivision, current guidance works)—bank consolidation, thresholds, cutwater, rock-fills, concrete blocks, stone masonry, and reinforced concrete (classification code 1.4.2.2.)—have a normal service life of 24–36 years. Under these circumstances, an average duration of 30 years was estimated as a normal running time of the torrent-correction works within the small Fâncel watershed.

*4.3. Estimation of the Value of the Income*

Considering that the water-management works of the small Fâncel watershed do not directly produce goods or services intended for commercialization, the value of the revenue was calculated by estimating the economic, social, environmental, etc., benefits on two distinct levels.

Firstly, the economic effects produced by reducing the transport and storage of alluvium was estimated. Secondly, the estimated value was calculated according to the potential damage that was avoided by the hydrological-planning works, such as damage to or destruction of some economic, utilitarian, or social objectives.

A. The economic effects produced by reducing the transport of alluvium

In order to estimate the economic effects caused by the diminution of the transport of alluvium, we considered that the area consolidated through the torrent-correction works within the small Fâncel watershed is 1129 $m^2$. The direct retention value was estimated for the entire duration as 1300 $m^3$, whereas the value of retention by consolidation was estimated to be 21,000 $m^3$. The total amount of alluvium retained was estimated to be 22,300 $m^3$ and took into account the volume of alluvium accumulated by both the longitudinal and transverse works [24].

As previously estimated, the unitary cost of the manual coarse alluvium extraction was RON 80/$m^3$ and the cost of the mechanized coarse alluvium extraction was updated to RON 69/$m^3$. A total of 50% is supposed to be extracted in a mechanized way and 50% manually, which implies an average cost of RON 75/$m^3$. The abovementioned costs were calculated at the rates used by the specialized companies in the region in 2018. Therefore, the costs that would have been necessary for the excavation of torrential alluvium would have been approximately RON 1,661,350 (Table 3).

**Table 3.** Estimated cost of sediment excavation.

| Name | U.M. | RC | RD |
|---|---|---|---|
| Volume of alluvium | $m^3$ | 21,000 | 1300 |
| Unitary cost | RON/$m^3$ | 74.5 | 74.5 |
| Estimated value | RON | 1,564,500 | 96,850 |
| Estimated total | RON | 1,661,350 | |

Notes: RC = retention by consolidation, RD = direct retention.

It can therefore be estimated that the economic effects produced by the reduction in the transport of alluvium accumulated by the torrent-correction works within the hydrographic small Fâncel watershed would have amounted to RON 1,661,350 if they had to be excavated in 2018. Considering the normal 30-year functioning duration, it can be estimated that the annual economy generated by the reduction in the transport of alluvium caused by the torrent-correction works within the small Fâncel watershed amounted to RON 55,378.33.

B. Estimation of the cost of potential damage avoided

In order to estimate the value of the potential damage that was avoided due to the torrent-correction works, the following assumption was taken into account as a starting

point: If the torrent-correction works in the small Fâncel watershed had not been carried out, a series of economic and social assets would have suffered, including the county road DJ153, the villages Dulcea and Brădeţelul, the related forestry and agricultural funds, and the bridges.

The impact class of the Fâncel watershed has been established according to STAS 5576-88, STAS 4273-83 [78,79]. The impact class is determined according to the vulnerability ranking of the objectives that are protected (Table 4).

**Table 4.** Social and economic objectives to protect from torrential events.

| No | Name of Drainage Watershed | Name of Protected Objective | Impact Class | Vulnerability Rating |
|----|---|---|---|---|
| 1 | Milestone river 57 | Forest road | IV | III |
|  |  | Forest fund | IV | II |
|  |  | Villages Dulcea, Brădeţelul | III | I |
| 2 | Milestone Pr. Zapodia Scurtă 85 | Forest road | IV | III |
|  |  | Forest fund | II | II |
|  |  | Villages Dulcea, Brădeţelul | III | II |
| 3 | Milestone river 47 | Forest road | IV | III |
|  |  | Forest fund | IV | II |
|  |  | Villages Dulcea, Brădeţelul | III | I |
|  |  | Agricultural fund | III | I |
| 4 | Milestone river 43 | Forest fund | III | II |
|  |  | Forest road | IV | II |
|  |  | Villages Dulcea, Brădeţelul | II | III |
|  |  | Agricultural fund | III | I |
| 5 | Milestone river 40 | Forest road | II | II |
|  |  | Forest fund | III | III |
|  |  | D.J.153 | II | II |
| 6 | Pr. Buneasa | Villages Dulcea, Brădeţelul | III | II |
|  |  | Forest fund | III | II |
| 7 | Milestone Pr. Porcul de Jos 22 | Forest road | IV | II |
|  |  | Villages Dulcea, Brădeţelul | II | III |
|  |  | Agricultural fund | III | I |
| 8 | Milestone Pr. Tarcea de Sus 80 | villages Dulcea, Brădeţelul | III | II |
|  |  | Forest fund | II | I |
|  |  | Forest road | III | II |
| 9 | Milestone Pr. 45 | Forest road | II | II |
|  |  | Forest fund | III | III |
|  |  | D.J.153 | II | II |
| 10 | Pr. Tabacu | Forest road | II | II |
|  |  | Forest fund | II | III |
|  |  | D.J.153 | II | II |

Note: Source: Preliminary studies to substantiate the need for hydrological works in the small hydrographic Fâncel watershed [55].

The degree of vulnerability is divided into three categories:

III—Very Vulnerable—the objective can be damaged in a proportion of more than 50% in the area of action of the waters, requiring dragging out mud, restoration, and consolidation.

II—Vulnerable—the objective can be damaged by 25–50% in the area of action of the waters, requiring dragging out mud, restoration, and consolidation.

I—Slightly Vulnerable—the objective can be damaged up to 25% in the area of action of the waters, requiring dragging out mud, restoration, and consolidation.

Forest roads are extremely vulnerable (Table 4), as their positions and their constructive features expose them to floodwaters. Riverbeds are unstable due to the transport of alluvium, passive deposits, and deep erosion. Moreover, as a result of undermining of riverbanks, there are collapses in the road slopes. Considering these characteristics, the road can be damaged by 40% in the water-action area, requiring consolidation and refurbishment [55].

The torrent-correction works in these pools led to the following functional capacities in the area: consolidation of 3 km of torrent riverbanks; consolidation of 2.0 ha of banks and slopes, alluvial deposits, and landings; groundwater retention of 1300 m$^3$ of alluvium; on-site consolidation of a volume of 21,000 m$^3$ of alluvium; protection of 1.7 km of main forest roads; protection of two poor footbridges; and protection of five tubular footbridges. The lack of these works would have led to a number of negative economic, social, and environmental impacts, both directly and indirectly.

The direct negative effects include blocking or damaging the communication ways with county road DJ153 (infrastructure, superstructure, and footbridge); destruction of or damage to the forest roads in the watersheds; clogging the footbridges and roads in the crossing area; flooding the properties in the water-action zone; shores and landslides, with direct consequences on the nearby forest fund (with trees and shrubs involved); alluvium on agricultural and forestlands and their removal from the economic circuit; clogging the riverbed of the Gurghiu River due to alluvium transport and deposits.

The indirect negative effects include additional costs for the unblocking, consolidation, or rehabilitation of the transport routes: the county road, forest roads, and communal roads; losses caused by traffic interruption; additional expenses for the restoration of individual households in the area; increase in the destructive potential of floods along the Gurghiu River and the Negru, Fâncel, and Fătăciunita streams; and additional costs for the restoration of the drainage section along the Gurghiu River.

These interventions are essential, as before their existence torrential floods have led to the decommissioning of forest roads for certain periods of time due to clogging or bridges being dislocating and roads being destroyed. This has occurred mainly in the sector where torrential formations were present, or downstream, where deep erosion has undermined the embankments or where deposits of alluvium have formed on bedsides diminished by leakage sections.

The impact of hydro-technical construction is a social priority and is manifested by increasing safety and trust among the population and the economic agents in the protected area, in connection with the best protection of their lives and property. According to experts [80–82], in the small Fâncel watershed area, the damage caused by floods can be significant, but the value of the potential damage has not been numerically quantified. As a result, it is very difficult to appreciate the amount of damage that has been avoided due to the fact that hydrological planning works were carried out in the small Fâncel watershed.

A prudent approach to potential damage is preferred. The estimated values are summarized in Table 5. Taking into consideration specific assumptions, all the objectives were identified based on the preliminary studies carried on in order to substantiate the need for torrent-correction works within the small Fâncel watershed. Once identified, the estimated value for each target was achieved through direct field visits and by consultation with experts in the field. For the buildings and the land, the unit value was estimated on the basis of the market survey at the minimum values recorded on the real-estate market in 2017 (valid for 2018), elaborated at the request of the National Union of Notaries [83]. For the forest roads and footbridges, the unit value was estimated based on the assessment of construction works specific to the reduction of torrential effects drafted in 2018. The estimations of the damage percentages were carried out based on the preliminary studies. The risk of a dimensioning flood was considered to be of 1 in 10 years, according to past registered events [84].

Based on the abovementioned calculations, an estimation of the revenue of the hydrological-improvement works within the small Fâncel watershed was therefore possible, and led to a total amount of RON 505,327.8/year, calculated as the amount of the annually avoided damage value (RON 449,949.5/year) and the value of the annual economic effects produced by the reduction in alluvium transport (RON 55,378.3/year) (Table 5).

**Table 5.** Evaluation of annual avoided damages.

| Name | Value of the Affected Area | | | | | VR | Damage Estimation | | | | | | TE |
| | Objectives | MU | Q | UV (RON) | TV (RON) | | Annual Floods | | | Drafting Floods | | | |
| | | | | | | | PA | NV | TV (RON) | PA | NV | TV (RON) | |
| Inside village | Buildings | m² | 1320 | 700 | 924,000 | III | 20% | 1 | 184,800 | 50% | 0.1 | 46,200 | 231,000 |
| | gardens, orchards | m² | 7183 | 6 | 43,098 | III | 20% | 1 | 8,619,6 | 50% | 0.1 | 2,154.9 | 10,774.5 |
| Forestry roads | Road trips | km | 1.7 | 450,000 | 765,000 | II | 15% | 1 | 114,750 | 40% | 0.1 | 30,600 | 145,350 |
| | Tile footbridge | piece | 2 | 80,000 | 160,000 | II | 15% | 1 | 24,000 | 40% | 0.1 | 6400 | 30,400 |
| | Tubular footbridge | piece | 5 | 23,000 | 115,000 | II | 15% | 1 | 17,250 | 40% | 0.1 | 4600 | 21,850 |
| Agricultural land | Agricultural culture | ha | 4.23 | 20,000 | 84,600 | I | 10% | 1 | 8460 | 25% | 0.1 | 2115 | 10,575 |
| | Total (RON) | | | | | | | | | | | | 449,949.5 |

Notes: N—number of floods; Q—quantity; UV—unitary value; TV—total value; VR—vulnerability rating; PA—percent of damage; TE—total estimated.

### 4.4. Economic-Efficiency Indicators

The analysis of the economic-efficiency indicators was carried on for each of the indicators considered necessary for expressing the efficiency of the investment.

1.  Investment (I)

The investment represents the value of the invested capital, increasing to RON 2,263,166 (in 2007) in the case of the hydrological facilities within the small Fâncel watershed. As already presented, the investment was discounted by means of a financial-discount rate (5%), and it was assessed as RON 1,323,226.29 at the value of 2018.

2.  Payback Period (P)

For the torrent-correction works within the small hydrographic Fâncel watershed, the period of the investment recovery was calculated to be 5 years, 0 months, and 15 days. Knowing that the normal service life for such a construction is between 24 and 36 years according to national law [39], and that for this particular investment the average duration was 30 years, the recovery period is almost six times less than the normal running time of torrent-correction works.

3.  Net Present Value (VNA)

Annual income was estimated at RON 505,327.8/year (totaling RON 15,159,834 for the entire period of 30 years), and the operating costs were zero. The used financial-discount rate, as mentioned above in this work, was 5% and the number of years was 30. The net present value of the torrent-correction works in the small Fâncel watershed was RON 5,612,730.67 in the updated value, which is positive and indicates that the investment is efficient.

4.  Internal Rate of Return (RIR)

The internal rate of return of the torrent-correction works within the small Fâncel watershed was calculated as 22%. This value is well above both the inflation rate (4.6% [46]) and the bank-interest rates on deposits (less than 3% [85]), which also proves the efficiency of the investment.

5.  Savings-to-Investment Ratio (IR)

The value of the savings-to-investment ratio of the torrent-correction works in the small Fâncel watershed was estimated to be 3.6%. This indicates that by the end of the normal operating period, the revenue would accumulate a value of 3.6 times the investment required and, as a result, each RON invested would generate a final potential profit (or costs avoided) of RON 3.6.

The centralization of the indicators of economic efficiency offers a synthetic and relevant image on the fact that the investment is effective in the analyzed conditions (Table 6). Therefore, it can be observed that in light of each of the indicators studied, the decision to make the investment was correct and able to generate positive effects in

the medium term. The present study adds new knowledge to the existing gap on the methodology used to estimate the efficiency of investments that produce public goods and services. The results offer arguments to assume that the decision to consider that particular investment as a good practice is fulfilled.

**Table 6.** Centralization of economic-efficiency indicators.

| Indicator | Measurement Unit | Value | Observation |
|---|---|---|---|
| Investment | RON | 2,263,166.00 | RON 1,323,226.29—at the value of year 2018, after the investment value was discounted |
| Payback Period | year | 5 years, 0 months, and 15 days | Six times shorter than normal service life Confirms the efficiency of the investment |
| Net Present Value | RON | 5,612,730.67 | is positive Confirms the efficiency of the investment |
| Internal Rate of Return | % | 22 | is much higher than the inflation rate and bank-interest rates on deposits Confirms the efficiency of the investment |
| Savings-to-Investment Ratio | No. | 3.6 | Each invested RON returns 3.6 RON net profit Confirms the efficiency of the investment |

Note: In Romania in 2018, the bank-interest rates on deposits were less than 3% (source: National Bank of Romania [85]) and the inflation rate was 4.6% (source: Romanian National Institute of Statistics [84]).

*4.5. Final Discussion*

Studies [86,87] reveal that anthropogenic activities directly influence channel morphology in more than 90% of the total watercourse, and sustainable land-use practice needs to observe the specific individualities of watershed to achieve effective flood-risk management [88]. The results of the study confirm the findings [89,90] that an appropriate approach for forest-area planning has to consider the socio-economic and ecological realities. For this reason and through an applicative interdisciplinary approach, better benefits are expected in order to solve the problems of small water resources [91]. The work includes specific approaches not just for forest sciences but for ecological, socioeconomic, and financial sciences as well.

A World Bank Policy Research Working Paper [92] concluded that there are analyses that have revealed that increasing losses from disasters is slower than increasing wealth. However, in many countries, rising incomes have also led to increased migration and investment in high-risk areas [93], and a simple model shows that this trend may dominate the effect of risk-reduction measures. However, this assessment is considered an underestimation, since it does not take into account the indirect losses caused by natural disasters, e.g., the loss of production resulting from the loss of assets. For large-scale disasters, indirect losses can be of the same magnitude as direct losses.

Data registered over the last 50 years observed that there were six major floods recorded in this hydrographic basin: in 1970 (May), 1975 (July), 1981 (March), 1995–1996 (December 1995–January 1996), 1998 (June), 2005 (August), and 2010 (July) [94]. The floods in 1970 were caused by exceeding the transport capacity of the bed; the increase in water volume was caused by spring floods due to snowmelt and flooding of other kinds. The causes of the flood in 1975 (July) were massive rainfall, exceeding the transport capacity of the bed, and flash floods. In 1981, the flooding was caused by rainfall, exceeding the transport capacity of the bed, and flash floods. Between December 1995 and January 1996, the floods produced in B.H. Mureș were due to precipitation in the form of rain and the melting of snow in the mountain areas on account of high temperatures (13–14 °C). The cause of the 1998 flood was the large amount of precipitation that fell between 11 and 20.06.1998. The floods of 2005 and 2010 were local torrential floods generated by precipitation in the form of local showers. Since 2012, the year the Fâncel watershed was finished, no major flood has been recorded in the area.

*4.6. Methodology Validity*

As mentioned, there is no widely accepted methodology for analyzing a watershed. Opinions of scholars oscillate from the idea of no necessity of cost–benefit analysis for investments that generate public goods and/or services, such as a watershed [54], to the strict advice of running a cost–benefit analysis for any investment in watersheds [37,39,40].

Several attempts to propose a methodology for a cost–benefit analysis for any investment in watersheds based on case studies are available for evaluation. Using similar indicators (net present value, cost–benefit ratio, and internal rate of return), a study carried on in Karnataka on a watershed-development project [95] proved that the benefits make it desirable. The proposed method uses estimations, accepting that some direct and indirect benefits were not included because information was not available, and that other indirect benefits, mostly of an environmental nature, were included. A similar study with comparable results was conducted on the Bichhiwada watershed in India [96], where the benefits were estimated in terms of agriculture-output increase. In addition, in the Chaudiere River watershed in Canada [97], a cost–benefit analysis was carried out based on an environmental case study for an agricultural watershed. The proposed method is based on a sensitivity analysis, aiming to determine the potential effect of certain parameters: profit, treatment costs for manure, and the probabilities of exceeding the targeted water-quality standards.

Another study on a number of watersheds, including small ones was carried out in Farta Woreda, South Gondar, Ethiopia [98], and incorporated the most important costs and returns extrapolated to the future through a cost–benefit analysis. The methods pointed out that the costs remained the same, whereas the benefits increased over time. In Hawai'i Island, United States of America (USA), the method of cost–benefit analysis for a watershed was used for studying the result of onsite conservation [99], focusing on the effects of evapotranspiration.

A simple model of cost-effectiveness analysis has also been proposed to address investments in low-impact development designated to reduce the effects of sewer overflows in urban watersheds referring to Gowanus Canal (Brooklyn, NY, USA) [100]. In addition, the Duck Creek watershed in Clark County, southern Nevada, USA [101] is a model for assessing the cost effectiveness of stormwater management in an urban watershed. The study concluded that, at the moment, the knowledge related to best practices for the management of stormwater in the desert/semi-desert of southwestern USA is not adequate, so it advocated for additional research. Additional studies aim to observe the effects of cost-effectiveness analysis of urban stormwater watersheds [102,103], pointing out the lack of difficulty to estimate intangible benefits and costs.

Each case-study analysis brings new and valuable knowledge but also accepts its limits. They encourage new research and new methodological approaches [104,105], as this study does, to increase the replicability potential.

This study reveals the fact that human intervention in forest-river basins should be an essential factor for assuring their resilience and sustainable development. This is consistent with other studies that analyzed the complexity of the connection between contradictory objectives, such as ecosystem conservation [106], or between economic advancement and the wellbeing of cultural ecosystems [107]. Although environmental factors seem to be obvious, as forests are part of nature, the social and economic factors are not so easy to analyze. It can also be concluded that river-basin management works contribute to the reduction of the destructive character of torrential events and, therefore, to the diminution of the flood risk to the dwellings and socio-economic objectives of the area.

## 5. Conclusions

The study's main objective was to observe whether investment in hydrographic watersheds could be used as a model of good practice for sustainable development. Based on the abovementioned results, it can be stated that investment in hydrographic watersheds can be disseminated as good practice. This affirmation is based on the fact that real-value costs were used and not economic ones. In addition, appreciating the positive effects on

the area due to the investment, it can be confirmed that it is a good model for sustainable development. The secondary objective was to draw attention to the importance of in-depth analysis of the economic efficiency of investments in watersheds, and this was emphasized throughout the entire study.

By centralizing the economic efficiency, indicators of the torrent-correction works within the small hydrographic Fâncel watershed provided a synthetic and relevant image of the investment as effective in the terms under review. Therefore, it can be concluded that the decision to make the investment was correct and able to generate positive effects in the medium term from the point of view of each of the studied indicators. In this research, the interventions to control negative effects of torrents were considered as investment, and they were economically analyzed by using cost–benefit analysis. The results were encouraging, proving that the effects, expressed in a financial manner, are positive. In that way, the findings are a starting point for setting up a framework of practical methodology for watershed management.

The study has its limitations, as it was carried out on a small watershed, which, of course, has its own specificities. Nevertheless, this kind of research is relevant when the model is extended to a provincial level on a larger scale or to a large watershed [62,108]. In addition, because of the inability to know precisely the numbers in all categories of costs and revenues, various methods were used to estimate them, so additional similar studies should be conducted whenever possible, and the efficiency of investments should be analyzed and compared.

**Author Contributions:** Conceptualization, I.D.A.; data curation, I.D.A.; formal analysis, F.H.A., I.T., M.I. and I.C.M.; investigation, I.D.A.; methodology, F.H.A., M.I., I.T. and I.C.M.; supervision, I.D.A. and M.D.; validation, F.H.A.; writing—original draft, I.D.A. and F.H.A. All authors have read and agreed to the published version of the manuscript.

**Funding:** This research received no external funding.

**Institutional Review Board Statement:** Not applicable.

**Informed Consent Statement:** Not applicable.

**Data Availability Statement:** No new data were created or analyzed in this study. Data sharing is not applicable to this article.

**Acknowledgments:** Part of the study was carried out as part of the doctoral thesis of Iulia Diana Arion, "Studies regarding the efficiency of hydrotechnical works from hydrographic Basin 10Gurghiu (Cercetări privind eficiența lucrărilor hidrotehnice din Bazinul Hidrografic al Gurghiului)," coordinated by Marcel Dirja, in the framework of the Doctoral School Agricultural Engineering Sciences of the University of Agricultural Sciences and Veterinary Medicine of Cluj-Napoca.

**Conflicts of Interest:** The authors declare no conflict of interest.

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
