# Peer review of "Investment in Forest Watershed—A Model of Good Practice for Sustainable Development of Ecosystems"

_water, doi:10.3390/w15040754_

Round 1
Reviewer 1 Report
The manuscript could be of interest but, despite other minor issues, in my opinion there is one major concern that prevents from drawing a clear opinion. From the text it seems the authors are presenting the cost-benefit for one project. If thet is the case the authors shoudl describe the project and present the cientific interest of that case and discuss the reproducibility and general interest.
Other minor changes are:
It looks some figures include words in romanian languaje.
I don´t think RON currency is meaningful enough for an international reader.
Author Response
Comments from Reviewer
The manuscript could be of interest but, despite other minor issues, in my opinion there is one major concern that prevents from drawing a clear opinion.
The authors really appreciate your time, your concern, and your dedication to offering us constructive suggestions to impro the article. We express our gratitude for all of these. Our point-to-point reply is provided in below.
Modifications to the manuscript have been highlighted in yellow.
From the text it seems the authors are presenting the cost-benefit for one project. If thet is the case the authors should describe the project and present the scientific interest of that case and discuss the reproducibility and general interest.
The sub-chapter “3.5. Methodology validity” was completed to compare the method with other studies that used cost-benefit analysis in different watersheds study cases in diverse countries, (lines 543-581)
3.5. Methodology validity
As mentioned, there is no widely accepted methodology for analyzing a watershed. Opinions of scholars oscillate from the idea of no necessity of cost-benefit analysis for investments that generate public goods and/or services – such as a watershed [54], to the strict advice of running a cost-benefit analysis for any investment in watershed [37, 39, 40].
There are available to evaluate several attempts to propose a methodology for a cost-benefit analysis for any investment in watersheds based on case studies. Using similar indicators (Net Present Value, Benefit-Cost Ratio and Internal Rate of Return) a study carried on in Karnataka on a watershed development project [95] proved that the benefits make it desirable. The proposed method uses estimations, accepting that as some direct and indirect benefits were not included, because information were not available, and other indirect benefits, mostly of an environmental nature, are included. Similar study, with comparative results, was done on Bichhiwada Watershed, India [96], where the benefits were estimated on agriculture output increase. Also, in Chaudiere River watershed, Canada [97], it was carried on a benefit-cost analysis based on an environmental case study for an agricultural watershed. The proposed method is based on a sensitivity analysis aiming to determine the potential effect of some parameters: profit, treatment costs for manure, and the probabilities of exceeding the targeted water quality standards.
Another study – on a number of watersheds, including small ones -, carried on in Farta Woreda, South Gondar, Ethiopia [98], incorporating incorporates the most important costs and returns extrapolated to the future through a cost-benefit analysis. The methods pointed out that the costs remain the same, while the benefits increase over time. In Hawai’i Island, United States of America (USA), the method of bene-fit-cost analysis of watershed was used for studying the result on site conservation [99], focusing on the effects of evapotranspiration.
There is, also, proposed a simple model of cost-effectiveness analysis for addressing the investments in low-impact development designated to reduce the effects of sewer overflows in urban watersheds, though referring to Gowanus Canal (Brooklyn, NY) USA [100]. Also, The Duck Creek watershed in Clark County, southern Neva-da, USA [101], a model for assessing the cost-effectiveness of stormwater management in an urban watershed. The study concluded that, at the moment, the knowledge related to stormwater best management practices in the desert/semi-desert southwest America is not adequate, so they advocate for additional research. Additional studies aim to observe the effects of cost-effectiveness analysis of urban stormwater watersheds [102, 103], pointing out the lacking of difficulty to estimate intangible benefits and costs.
Each case study analysis brings new and valuable knowledge but also accepts its limits. They are encouraging new research and new methodological approaches, as this study is doing, for increasing the replicability potential.
Other minor changes are:
It looks some figures include words in Romanian language.
Thank you for pointing this out. All the information were translated into English Language, so to be understand by a broader audience. See Figure 1 and Figure 2
I don´t think RON currency is meaningful enough for an international reader.
We are understanding these issues, and as we confront any time, we use national data. As the exchange rate of US/RON is quite volatile, so the data should be understood at the. But, for a better understanding of the values, we mentioned the average exchange rate at the moment of the analysis (4.6540 RON for 1 EURO (see line 320)
Reviewer 2 Report
A moderate revision is required to enhance the quality of this study. My concerns were listed as follows,
1) The abstract must be better organized to elaborate the scientific importance and soundness of this study, particularly to highlight the innovations in methodology and new findings of this study. Besides, some quantitative results should also be added.
2) In section 1, what are the researching gap(s) in previous studies?
3) I suggest the elevation or DEM data should be incorporated in Figure 1.
4) In Table 4, I suggest that analysis of variance (ANOVA) or non-parametric ANOVA methods should be used to validate the differing classes between these rivers..
5) Innovations and limitations of this study should be highlighted. And what should be done to enhance further studies to put this researching domain forward.
Author Response
Comments from Reviewer
A moderate revision is required to enhance the quality of this study. My concerns were listed as follows,
The authors really appreciate your time, your concern, and your dedication to offering us constructive suggestions to impro the article. We express our gratitude for all of these. Our point-to-point reply is provided in below.
Modifications to the manuscript have been highlighted in yellow.
1) The abstract must be better organized to elaborate the scientific importance and soundness of this study, particularly to highlight the innovations in methodology and new findings of this study. Besides, some quantitative results should also be added.
The Abstract was reconsidered, so to answer the mentioned requirements. Quantitative results were, also, added.
Abstract: Estimating the efficiency of the investments that generate public benefits is not an easy task to do, as there are various methods proposed for evaluating the value of public goods and services. The present study proposes a cost-benefit methodology adapted to a real-value cost concept for estimating the efficiency of the investment in the hydrographic watershed Fâncel, in the Centre of Romania, using a set of 5 indicators. The results proved that the Investment 1,323,226.29 RON, at the value of 2018, would be paid in 5 years, 0 months and 15 days. The estimated income is 505,327.8 RON/year, so for a 5% financial discount rate and an estimated period of use of number 30 years, the net Present value is 5,612,730.67 RON. The internal rate of return was calculated as 22%, while the value of the savings to invest ratio is 3.6%. The main findings of the analysis of the economic efficiency indicators offer a synthetic and relevant image of the fact that the investment has proven to be effective under the analysed conditions. The results offer arguments to assume that the decision to consider that particular investment a good practice is fulfilled.
2) In section 1, what are the researching gap(s) in previous studies?
We emphasized the scientific gap that the manuscript is addressing, mainly the inconsistent methodology used for cost-benefit analysis. The manuscript is proposing a one based on the real-value cost concept that is, also used by The World Bank Group for public goods generating investments (like our case study is). The details can be seen in the lines 151-161
The main objective of the work is to evaluate the investment in the hydrographic watershed. The evaluation could be used as a model of good practice for sustainable development, as an integral part of the water-forest integrated strategy for the area. The study is addressing a knowledge gap in the methodology of effective investments in infrastructures that deal with natural factors, offering a new perspective on the subject. For this, the economic efficiency of hydrographic watershed, based on adapted cost-benefit methodology with real-value cost concept, was estimated. The work is multifaceted, based on the complexity of the estimation of the potential cost and revenues, which implies expressing units of the various types of non-financial goods in economic (e.g. environmental, social and not only). The secondary objective is to draw attention to the importance of an in-depth analysis of the economic efficiency of investments in watersheds.
On the sub-chapter 2.2. Adopted method, the previous relevant studies were examined, to answer to the identified gap.
2.2. Adopted method
In order to determine the choices that maximize the benefits and suppose a monetary expression of all direct, indirect and intangible effects, it is necessary to apply the cost-benefit analysis (CBA), which supposes the comparison of the cost of a project with the benefits which it can bring. The used methodology is an adaptation of the recommendations of The World Bank Group [56] which emphasize the importance of economic and financial analysis on the process-making for watershed investment. It takes into consideration of inputs and outputs, including externalities and “cost of inaction” [57] expressed as real-value costs.
Studies proved that a compromise between the cost of an investment and the results should be achieved, expressed both qualitatively and quantitatively [58]. Thus, the situation after the investment should be compared with the one when the investment was not made. So, the correct setting of the real level for comparing reference accounts for the key factor to express the cost-benefit for an investment in ecosystems [59]. There were various researches studying [60, 61] the effects of using the concept of economic efficiency of the investments in torrential watershed development, created to regenerate the natural potential. They expose the differences between the cases with and without investment in monetary units. In practice, there are more ways of analysing both economic costs and benefits in the process of decision making for watershed investments and maintenance [62], adapted to the local specificity.
It can be observed that the methodology is so differently approached. There are scholars [63] who advocate that cost-effective strategy, and mainstreaming investments in watershed services, require the development of practical tools for costs and benefits measurement. Consequently, the dissemination of good practices should be replicable both institutionally and financially.
3) I suggest the elevation or DEM data should be incorporated in Figure 1.
Thank you for this observation. On figure 1 the map was completed, so the elevation of the studied area to be easily observed.
4) In Table 4, I suggest that analysis of variance (ANOVA) or non-parametric ANOVA methods should be used to validate the differing classes between these rivers.
We appreciate your comment. As ANOVA shows if there are any statistical differences between the means of three or more independent groups. However, in this case, there is no mean that might be compared, as the values are based on the evaluation of the risk of vulnerability, so calculating the mean of them has no meaning. We assume that, most probably, we did not explain enough of that value on the table. So, we better evidence that on lines 393-403
The impact class of Watershed Fâncel has been established according to STAS 5576-88, STAS 4273-83 [78], [79]. The impact class is determined according to the vulnerability ranking of the objectives that are protected (Table 4).
The degree of vulnerability is divided into 3 categories:
III - very vulnerable - the objective can be damaged in a proportion of more than 50%, in the area of action of the waters, requiring dragging out of mud, restorations and consolidations.
II – vulnerable - the objective can be damaged by 25-50%, in the area of action of the waters, requiring dragging out of mud, restorations and consolidations.
I- a little vulnerable - the objective can be damaged up to 25%, in the area of action of the waters, requiring dragging out of mud, restorations and consolidations.
5) Innovations and limitations of this study should be highlighted. And what should be done to enhance further studies to put this researching domain forward.
The innovation is emphasized as addressing the scientific gap that the manuscript is addressing, mainly the inconsistent methodology used for cost-benefit analysis. The manuscript is proposing one based on the real-value cost concept which is, also used by The World Bank Group for public goods generating investments (like our case study is). The details can be seen on lines 393-403, where the scientific gap is introduced.
Also, the sub-chapter “3.5. Methodology validity” was completed to compare the method with other studies that used cost-benefit analysis (lines 543-581) in different watershed study cases in diverse countries,.
3.5. Methodology validity
As mentioned, there is no widely accepted methodology for analyzing a watershed. Opinions of scholars oscillate from the idea of no necessity of cost-benefit analysis for investments that generate public goods and/or services – such as a watershed [54], to the strict advice of running a cost-benefit analysis for any investment in watershed [37, 39, 40].
There are available to evaluate several attempts to propose a methodology for a cost-benefit analysis for any investment in watersheds based on case studies. Using similar indicators (Net Present Value, Benefit-Cost Ratio and Internal Rate of Return) a study carried on in Karnataka on a watershed development project [95] proved that the benefits make it desirable. The proposed method uses estimations, accepting that as some direct and indirect benefits were not included, because information were not available, and other indirect benefits, mostly of an environmental nature, are included. Similar study, with comparative results, was done on Bichhiwada Watershed, India [96], where the benefits were estimated on agriculture output increase. Also, in Chaudiere River watershed, Canada [97], it was carried on a benefit-cost analysis based on an environmental case study for an agricultural watershed. The proposed method is based on a sensitivity analysis aiming to determine the potential effect of some parameters: profit, treatment costs for manure, and the probabilities of exceeding the targeted water quality standards.
Another study – on a number of watersheds, including small ones -, carried on in Farta Woreda, South Gondar, Ethiopia [98], incorporating incorporates the most important costs and returns extrapolated to the future through a cost-benefit analysis. The methods pointed out that the costs remain the same, while the benefits increase over time. In Hawai’i Island, United States of America (USA), the method of bene-fit-cost analysis of watershed was used for studying the result on site conservation [99], focusing on the effects of evapotranspiration.
There is, also, proposed a simple model of cost-effectiveness analysis for addressing the investments in low-impact development designated to reduce the effects of sewer overflows in urban watersheds, though referring to Gowanus Canal (Brooklyn, NY) USA [100]. Also, The Duck Creek watershed in Clark County, southern Neva-da, USA [101], a model for assessing the cost-effectiveness of stormwater management in an urban watershed. The study concluded that, at the moment, the knowledge related to stormwater best management practices in the desert/semi-desert southwest America is not adequate, so they advocate for additional research. Additional studies aim to observe the effects of cost-effectiveness analysis of urban stormwater watersheds [102, 103], pointing out the lacking of difficulty to estimate intangible benefits and costs.
Each case study analysis brings new and valuable knowledge but also accepts its limits. They are encouraging new research and new methodological approaches, as this study is doing, for increasing the replicability potential.
We emphasized the limits of our study in the Conclusion part. The details can be seen in the lines 613-619
The study has its limitation as it was carried on for small watersheds which, of course, has its own specificities. Nevertheless, that kind of research is relevant when the model is extended to a provincial level on a larger scale or to a large watershed [104, 105]. Also because of the inability to know precisely the amount of all categories of costs and revenues, various methods were used to estimate them, so further similar studies should be conducted whenever possible, and the efficiency of investments should be analysed and compared.
Reviewer 3 Report
The work is interesting, has a clear degree of originality, and is appropriate for publication in the journal after performing a major and very careful revision. Nevertheless, it needs some further improvements. In general, there are still some occasional grammar errors throughout the manuscript, especially the article "the," "a," and "an" are missing in many places; please make spellchecking in addition to these minor issues. The reviewer has listed some specific comments that might help the authors further enhance the manuscript's quality.
- Specific Comments
· Overall, the Abstract section is not giving any information about methodology, results, conclusion, and recommendations as it should be with clear. I suggest the authors to remove generic lines and present the strong statements and novelty of article. The abstract written by qualitative sentences. It is need to modify and rewrite based on the most important quantity results from this research. The abstract should be redesigned. You should avoid using acronyms in the abstract and insert the work's main conclusion.
· You have used many abbreviations in the text. From this perspective, an Index of Notations and Abbreviations would be beneficial for a better understanding of the proposed work. Furthermore, please check carefully if all the abbreviations and notations considered in work are explained for the first time when they are used, even if these are considered trivial by the authors. The paper should be accessible to a wide audience. Furthermore, it will make sense to include also the notations in this index.
- The objectives should be more explicitly stated.
- The Introduction section must be written on more quality way. The research gap should be delivered on more clear way with directed necessity for the conducted research work.
- Please elaborate on the introduction section. The following literature may be helpful in this regard: << Small hydropower plants proliferation and fluvial ecosystem conservation nexus>>, << Water-energy-ecosystem nexus: Balancing competing interests at a run-of-river hydropower plant coupling a hydrologic–ecohydraulic approach >>, << Small Hydropower Plants’ Impacts on the Ecological Status Indicators of Urban Rivers>>, << A new approach to mapping cultural ecosystem services>>, you may consider additional references as well.
- What is the novelty of this work?
- It is better to improve your contributions which are not so clear to show the advantage of
your work.
· The novelty of this work must be clearly addressed and discussed in Introduction section.
- The methodology limitation should be mentioned.
Many equations are presented in the paper, and most look OK. However, please check carefully whether all equations are necessary and whether the quantities involved are properly explained. Also, some equations need references.
- Results
- This section is well written.
- Discussion
- Overall, the discussion part is weak. The Discussion should summarize the manuscript's main finding(s) in the context of the broader scientific literature and address any study limitations or results that conflict with other published work.
- Conclusion
- Some future works should be added to your conclusion. Please elaborate it a bit more.
Author Response
Comments from Reviewer
The abstract must be better organized to elaborate the scientific importance and soundness of this study, particularly to highlight the innovations in methodology and new findings of this study. Besides, some quantitative results should also be added.
The Abstract was reconsidered, so to answer the mentioned requirements. Quantitative results were, also, added.
Abstract: Estimating the efficiency of the investments that generate public benefits is not an easy task to do, as there are various methods proposed for evaluating the value of public goods and services. The present study proposes a cost-benefit methodology adapted to a real-value cost concept for estimating the efficiency of the investment in the hydrographic watershed Fâncel, in the Centre of Romania, using a set of 5 indicators. The results proved that the Investment 1,323,226.29 RON, at the value of 2018, would be paid in 5 years, 0 months and 15 days. The estimated income is 505,327.8 RON/year, so for a 5% financial discount rate and an estimated period of use of number 30 years, the net Present value is 5,612,730.67 RON. The internal rate of return was calculated as 22%, while the value of the savings to invest ratio is 3.6%. The main findings of the analysis of the economic efficiency indicators offer a synthetic and relevant image of the fact that the investment has proven to be effective under the analysed conditions. The results offer arguments to assume that the decision to consider that particular investment a good practice is fulfilled.
In general, there are still some occasional grammar errors throughout the manuscript, especially the article "the," "a," and "an" are missing in many places; please make spellchecking in addition to these minor issues.
We required the support of a specialised external company for supporting us, so the manuscript has been thoroughly reviewed and edited with regard to the English language use, word order and spelling, as well as special language and terminology used.
The reviewer has listed some specific comments that might help the authors further enhance the manuscript's quality.
Specific Comments
- Overall, the Abstract section is not giving any information about methodology, results, conclusion, and recommendations as it should be with clear. I suggest the authors to remove generic lines and present the strong statements and novelty of article. The abstract written by qualitative sentences. It is need to modify and rewrite based on the most important quantity results from this research. The abstract should be redesigned. You should avoid using acronyms in the abstract and insert the work's main conclusions
The Abstract was reconsidered, so to answer the mentioned requirements. Quantitative results were, also, added.
Abstract: Estimating the efficiency of the investments that generate public benefits is not an easy task to do, as there are various methods proposed for evaluating the value of public goods and services. The present study proposes a cost-benefit methodology adapted to a real-value cost concept for estimating the efficiency of the investment in the hydrographic watershed Fâncel, in the Centre of Romania, using set of 5 indicators. The results proved that the Investment 1,323,226.29 RON, at the value of 2018, would be paid in 5 years, 0 months and 15 days. The estimated income is 505,327.8 RON/year, so for a 5% financial discount rate and an estimated period of used of number 30 years, the net Present value is 5,612,730.67 RON. The internal rate of return was calculated as 22%, while the value of the savings to invest ratio is 3.6%. The main findings of the analysis of the economic efficiency indicators offer a synthetic and relevant image of the fact that the investment has proven to be effective under the analysed conditions. The results offer arguments to assume that the decision to consider that particular investment a good practice is fulfilled.
- You have used many abbreviations in the text. From this perspective, an Index of Notations and Abbreviations would be beneficial for a better understanding of the proposed work. Furthermore, please check carefully if all the abbreviations and notations considered in work are explained for the first time when they are used, even if these are considered trivial by the authors. The paper should be accessible to a wide audience. Furthermore, it will make sense to include also the notations in this index.
An Index of Notations and Abbreviations was added as a supplementary Annex, so as to be easier understood by the readers. All used abbreviations and notations were checked to be explained at their first use.
- The objectives should be more explicitly stated. The Introduction section must be written on more quality way. The research gap should be delivered on more clear way with directed necessity for the conducted research work.
We emphasized the objectives of the study, as resulted from the identified scientific gap that the manuscript is addressing, mainly the inconsistent methodology used for cost-benefit analysis. The manuscript is proposing one based on the real-value cost concept which is, also used by The World Bank Group for public goods generating investments (like our case study is). The details can be seen in lines 151-161.
The main objective of the work is to evaluate the investment in the hydrographic watershed. The evaluation could be used as a model of good practice for sustainable development, as an integral part of the water-forest integrated strategy for the area. The study is addressing a knowledge gap in the methodology of effective investments in infrastructures that deal with natural factors, offering a new perspective on the subject. For this, the economic efficiency of the hydrographic watershed, based on an adapted cost-benefit methodology with a real-value cost concept, was estimated. The work is multifaceted, based on the complexity of the estimation of the potential cost and revenues, which implies expressing units of the various types of non-financial goods in economic (e.g. environmental, social, and not only). The secondary objective is to draw attention to the importance of an in-depth analysis of the economic efficiency of investments in watersheds.
- Please elaborate on the introduction section. The following literature may be helpful in this regard: << Small hydropower plants proliferation and fluvial ecosystem conservation nexus>>, << Water-energy-ecosystem nexus: Balancing competing interests at a run-of-river hydropower plant coupling a hydrologic–ecohydraulic approach >>, << Small Hydropower Plants’ Impacts on the Ecological Status Indicators of Urban Rivers>>, << A new approach to mapping cultural ecosystem services>>, you may consider additional references as well.
Thank you for pointing out these resourceful references. Each and any of them are interesting for the aim of our manuscript. We included them in the References lists (positions) and we consider them as being relevant for the discussion and/or conclusion part, which, is also, required, as you proposed. We introduced references 95, 105, 106, 107.
- What is the novelty of this work? It is better to improve your contributions which are not so clear to show the advantage of your work. The novelty of this work must be clearly addressed and discussed in Introduction section.
The innovation is emphasized as addressing the scientific gap that the manuscript is addressing, mainly the inconsistent methodology used for cost-benefit analysis. The manuscript is proposing one based on the real-value cost concept which is, also used by The World Bank Group for public goods generating investments (like our case study is). The details can be seen on lines 393-403, where the scientific gap is introduced.
The methodology limitation should be mentioned.
The limits of the method we proposed were mentioned of a dedicated part, meaning sub-chapter on the discussion part, the sub-chapter 3.5. Methodology validity. (see lines 543-581)
3.5. Methodology validity
As mentioned, there is no widely accepted methodology for analyzing a watershed. Opinions of scholars oscillate from the idea of no necessity of cost-benefit analysis for investments that generate public goods and/or services – such as a watershed [54], to the strict advice of running a cost-benefit analysis for any investment in watershed [37, 39, 40].
There are available to evaluate several attempts to propose a methodology for a cost-benefit analysis for any investment in watersheds based on case studies. Using similar indicators (Net Present Value, Benefit-Cost Ratio and Internal Rate of Return) a study carried on in Karnataka on a watershed development project [95] proved that the benefits make it desirable. The proposed method uses estimations, accepting that as some direct and indirect benefits were not included, because the information was not available, and other indirect benefits, mostly of an environmental nature, are included. A similar study, with comparative results, was done in Bichhiwada Watershed, India [96], where the benefits were estimated on agriculture output increase. Also, in Chaudiere River watershed, Canada [97], it was carried on a benefit-cost analysis based on an environmental case study for an agricultural watershed. The proposed method is based on a sensitivity analysis aiming to determine the potential effect of some parameters: profit, treatment costs for manure, and the probabilities of exceeding the targeted water quality standards.
Another study – on a number of watersheds, including small ones -, carried on in Farta Woreda, South Gondar, Ethiopia [98], incorporating incorporates the most important costs and returns extrapolated to the future through a cost-benefit analysis. The methods pointed out that the costs remain the same, while the benefits increase over time. In Hawai’i Island, United States of America (USA), the method of bene-fit-cost analysis of watershed was used for studying the result on site conservation [99], focusing on the effects of evapotranspiration.
There is, also, proposed a simple model of cost-effectiveness analysis for addressing the investments in low-impact development designated to reduce the effects of sewer overflows in urban watersheds, though referring to Gowanus Canal (Brooklyn, NY) USA [100]. Also, The Duck Creek watershed in Clark County, southern Neva-da, USA [101], a model for assessing the cost-effectiveness of stormwater management in an urban watershed. The study concluded that, at the moment, the knowledge related to stormwater best management practices in the desert/semi-desert southwest America is not adequate, so they advocate for additional research. Additional studies aim to observe the effects of cost-effectiveness analysis of urban stormwater watersheds [102, 103], pointing out the lacking of difficulty to estimate intangible benefits and costs.
Each case study analysis brings new and valuable knowledge but also accepts its limits. They are encouraging new research and new methodological approaches, as this study is doing, for increasing the replicability potential.
- Many equations are presented in the paper, and most look OK. However, please check carefully whether all equations are necessary and whether the quantities involved are properly explained. Also, some equations need references.
Each and any of the equations and formulas theoretically presented on the mythology part was used, after all, on the results part to obtain the outputs. Scientifical rigor requires to be introduced so as to be replicable.
We assumed that we were not clear enough mentioning that all the indicators used are according to the national guidelines for the cost-benefit analysis of the projects financed from the structural instruments, elaborated by the Authority for the Coordination of Structural Instruments within the Ministry of Economy and Finance (reference 65). We thank you for pointing out that mistake, so corrected that omission by mentioning the source in line 259.
- Results. This section is well written.
Thank you for your consideration.
- Discussion. Overall, the discussion part is weak. The Discussion should summarize the manuscript's main finding(s) in the context of the broader scientific literature and address any study limitations or results that conflict with other published work.
We further improved that part. We also consulted and added new references to sustain the findings in the context of the broader scientific literature, focussing on the most important part of the discussion, the one that is related to the validity of the used methodology. Please see sub-chapter 3.5. Methodology validity. (lines 543-581) .
We emphasized the limits of our study in the Conclusion part. The details can be seen in the lines 613-619
The study has its limitation as it was carried on for small watersheds which, of course, has its own specificities. Nevertheless, that kind of research is relevant when the model is extended to a provincial level on a larger scale or to a large watershed [104, 105]. Also because of the inability to know precisely the amount of all categories of costs and revenues, various methods were used to estimate them, so further similar studies should be conducted whenever possible, and the efficiency of investments should be analysed and compared.
- Conclusion. Some future works should be added to your conclusion. Please elaborate it a bit more.
We further improved that part. We also consulted and added new references to sustain the findings in the context of the broader scientific literature (References 105, 106, 107). Lines 592-607
This study reveals the fact that human interventions in forest river basins should be essential factor for assuring their resilience and sustainable development. This is consistent with other studies that analysed of the complexity of the connexion between contradictory objectives, like production ecosystem conservation [106], or between economic advance and the wellbeing of the cultural ecosystems [107]. While environmental factors seem to be obvious, as forest is part of the nature, the social and economic factors are not so easy to analyse. It can be concluded, also, that the river basin management works contribute to the reduction of the destructive torrential events destructive character and, therefore, to the diminution of the flooding risk on the dwellings and socio-economic objectives of the area.
By centralizing the economic efficiency, indicators of the torrent correction work within the hydrographic small watershed Fâncel provide a synthetic and relevant image of the investment as an effective investment in the terms under review. Therefore, it can be concluded by saying that the decision to achieve the investment is correct and able to generate positive effects in the medium term from the point of view of each of the studied indicators.
Round 2
Reviewer 1 Report
The authors have addressed my previous suggestions.
Author Response
The authors really appreciate your time, your concern, and your dedication to offering us constructive suggestions to impro the article. We express our gratitude for all of these.
Modifications to the manuscript have been highlighted in yellow.
Reviewer 2 Report
I think this manuscript can be considered for publication in its present form. Congratulations to the authors!
Author Response

(The authors gave the same response as above.)
